# Special temperatures in frustrated ferromagnets

L. Bovo [1,2], M. Twengström [3], O.A. Petrenko[4], T. Fennell[5], M.J.P. Gingras[6,7,8], S.T. Bramwell[1] & P. Henelius[3]

The description and detection of unconventional magnetic states, such as spin liquids, is a recurring topic in condensed matter physics. While much of the efforts have traditionally been directed at geometrically frustrated antiferromagnets, recent studies reveal that systems featuring competing antiferromagnetic and ferromagnetic interactions are also promising candidate materials. We find that this competition leads to the notion of special temperatures, analogous to those of gases, at which the competing interactions balance, and the system is quasi-ideal. Although induced by weak perturbing interactions, these special temperatures are surprisingly high and constitute an accessible experimental diagnostic of eventual order or spin-liquid properties. The well characterised Hamiltonian and extended low-temperature susceptibility measurement of the canonical frustrated ferromagnet $Dy_2Ti_2O_7$ enables us to formulate both a phenomenological and microscopic theory of special temperatures for magnets. Other members of this class of magnets include kapellasite $Cu_3Zn(OH)_6Cl_2$ and the spinel $GeCo_2O_4$.

[1] London Centre for Nanotechnology and Department of Physics and Astronomy, University College London, 17-19 Gordon Street, London WC1H OAH, UK. [2] Department of Innovation and Enterprise, University College London, 90 Tottenham Court Rd, Fitzrovia, London W1T 4TJ, UK. [3] Department of Physics, Royal Institute of Technology, SE-106 91 Stockholm, Sweden. [4] Department of Physics, University of Warwick, Coventry CV4 7AL, UK. [5] Laboratory for Neutron Scattering and Imaging, Paul Scherrer Institut, 5232 Villigen PSI, Switzerland. [6] Department of Physics and Astronomy, University of Waterloo, Waterloo, ON N2L 3G1, Canada. [7] Canadian Institute for Advanced Research, 180 Dundas St. W., Toronto, ON M5G 1Z8, Canada. [8] Perimeter Institute for Theoretical Physics, 31 Caroline St. N., Waterloo, ON N2L 2Y5, Canada. Correspondence and requests for materials should be addressed to M.Töm. (email: mikaeltw@kth.se)

Models of magnetic frustration on regular lattices have naturally tended to focus on the case where there is a single interaction of one sign that is frustrated by the lattice geometry. Examples include the triangular or kagome lattice antiferromagnets[1–3], the pyrochlore Heisenberg antiferromagnet[4, 5] and spin ice in the near-neighbour approximation, a frustrated ferromagnet[6]. While there are many real materials that roughly approximate these ideal models[7–9], the nature of real magnetic interactions is such that a competition between antiferromagnetic (AF) and ferromagnetic (FM) interactions is commonly encountered. This arises because the superexchange interaction is fundamentally the difference between two large numbers—an AF and a FM part—and small differences in orbital overlap can tip it in one direction or the other[10]. Also, the dipole–dipole interaction, which is important in rare earth systems, has a sign that depends sensitively on direction. Hence, while near-neighbour interactions are of one sign, further neighbour interactions may be of the opposite sign. In the context of a geometrically frustrated lattice, it has recently been recognised that this competition can produce some interesting effects, including spin-liquid behaviour[11, 12], magnetic fragmentation[13], competing ground states[14, 15] and spin glass physics[16]. Many of these materials show a conspicuous broad peak in $\chi T/C$ (where $\chi$ is the magnetic susceptibility and $C$ the Curie parameter), which is the analogue of the product $pV/nRT$ in gas thermodynamics, and the focus of this work.

Classical gases exhibit a number of temperature values that signal transitions between contrasting physical properties[17, 18]. We label these 'special temperatures' to emphasise that they do not simply reflect characteristic or typical energy scales. They include the Boyle and Joule temperatures, with the most notable one being perhaps the Joule–Thomson (or inversion) temperature, $T_{JT}$, below which a gas may be liquefied by the Linde–Hampson process, which underpins a vast low-temperature technology. A particularly remarkable aspect of $T_{JT}$ is how large it is. For example, for nitrogen, $T_{JT} = 621$ K, even though the thermally averaged potential energy that gives rise to the finite $T_{JT}$ accounts for only about one thousandth of the internal energy of the system. In terms of the van der Waals equation of state, $p = \frac{RT}{V/n-b} - \frac{an^2}{V^2}$, $T_{JT} = 2a/(bR) \approx 27T_c/4$. Hence, $T_{JT}$ presents a surprising signature of the eventual liquid state in a temperature regime where at first sight, the intermolecular interactions are negligible. Until now, the magnetic analogies of these special temperatures foreshadowing phenomena at much lower temperature appear not to have been noticed.

In this work we put forward the concept of a class of 'inverting' frustrated ferromagnets, which exhibit a maximum in $\chi T/C$ as a function of temperature. In strong analogy with the theory of classical gases, we identify the peak in $\chi T/C$ with a magnetic Joule temperature, $T_J$, where the system is quasi-ideal, and the internal energy $U$ is independent of the magnetisation $M$, $(\partial U/\partial M)_T = 0$. The Joule temperature marks the onset of the low-temperature antiferromagnetic correlations. In addition, we identify and define a magnetic Boyle temperature, $T_B$, at which point $\chi T/C = 1$, and the incipient ferromagnetic correlations cross over to antiferromagnetic at low temperature. So while the magnitude of $\chi T/C$ can be used to classify magnets as ferromagnets ($\chi T/C > 1$) or antiferromagnets ($\chi T/C < 1$)[19, 20], we here focus on the special temperature values (points) of $\chi T/C$.

## Results

**Spin ice as a model inverting magnet.** Theoretically, the physics of special temperatures is hard to expose computationally in quantum spin systems due to the sign problem[21]. For frustrated systems with strong FM interactions, a major challenge is to

control demagnetising effects[22]. This makes the canonical frustrated ferromagnet spin ice[6, 7, 23–27] $Dy_2Ti_2O_7$ a natural starting point to explore the physics of competing FM–AF interactions. In this material, near-neighbour dipolar and exchange interactions average to a ferromagnetic coupling and a mapping to Pauling's model of water ice. However, further neighbour exchange and direction-dependent dipolar interactions provide competing couplings of opposite sign. By measuring the DC bulk susceptibility to lower temperatures than previously reported and using carefully crafted defect-free spherical single-crystal samples, which enables full control of demagnetising issues[22], we are able to identify the special temperatures in this well-studied material. $Dy_2Ti_2O_7$ lends itself naturally to this study as it stays close to the ideal paramagnetic limit ($\chi T/C = 1$) over a broad temperature range and remains a paramagnet well below its Curie–Weiss temperature on account of its high degree of frustration.

Thanks to the availability of a well-characterised Hamiltonian for $Dy_2Ti_2O_7$ (refs.[15, 28]), we are able to formulate a phenomenological model of the susceptibility which exposes the mechanism that induces the special temperatures and elevates the effects of minute-frustrated exchange interactions to surprisingly high temperature in this dipolar-coupled material. Furthermore, through an explicit numerical decomposition of the microscopic Hamiltonian, we demonstrate that these special temperatures, and eventual antiferromagnetic ordering, are caused by the weak quadrupolar corrections to the primary monopolar (dumbbell) Hamiltonian. Our study therefore establishes $\chi T/C$ as a measure of weak interaction parameters, which are otherwise difficult to access experimentally. From another broad context, the low-temperature susceptibility of spin ice is of particular interest in relation to 'topological sector fluctuations' of the harmonic component of the magnetisation[29, 30]. The analogy with the non-ideal gas allows an interpretation of the new experimental features in the magnetic susceptibility reported in this study, and have an appreciable impact on the interesting properties of spin ice—its residual entropy[7], magnetic monopoles[26] and Coulomb phase[27]—as discussed below.

**Experimental determination of the magnetic susceptibility.** A sphere of diameter 4 mm was commercially hand-cut from a larger crystal of $Dy_2Ti_2O_7$ (see refs.[30, 31]). Experimental conditions were carefully controlled to minimise measurement errors; see the Methods section. The experimental susceptibility of the sphere, $\chi_{exp}$, was determined from measurements of the magnetic moment, with a subsequent demagnetising correction to obtain the shape-independent intrinsic susceptibility, $\chi_{int}$,

$$\frac{1}{\chi_{int}} = \frac{1}{\chi_{exp}} - N, \tag{1}$$

using the exact result $N = 1/3$ for a sphere[22, 32].

From now on, we shall focus our discussion on the intrinsic susceptibility and suppress the 'int' subscript. The experimental measurement results are shown in Fig. 1. The Curie parameter is given by $C = \frac{N\mu_0\mu^2}{3Vk_B} = 3.92$ K for $Dy_2Ti_2O_7$, where $N/V$ is the ion density, and $\mu$ is the magnetic moment[30]. Our current measurements extend the earlier ones[30] (where the lowest temperature was 2 K), down to 0.5 K. The extended temperature range reveals the important physical phenomena that are the focus of the present study namely, a peak in $\chi T/C$ at $T_J \approx 2.2$ K, and a 'transition' from $\chi T/C > 1$ to $\chi T/C < 1$ at $T_B \approx 0.57$ K. These define the magnetic Joule and Boyle temperatures, respectively, as explained below. Alternatively, susceptibility measurements are often displayed as $1/\chi$ versus $T$. In the inset of Fig. 1, we show $C/\chi$ versus $T$, and note that the gradient at $T = T_J$ intersects the origin,

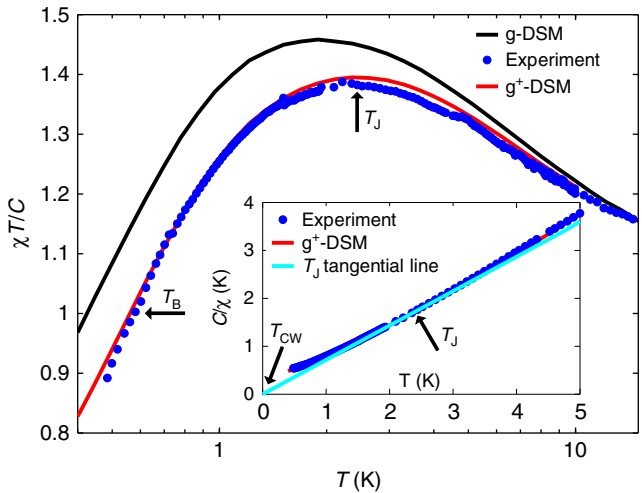

**Fig. 1** Magnetic special temperatures in spin ice, Dy$_2$Ti$_2$O$_7$. Experimental susceptibility $\chi T/C$ in blue, with arrows indicating the special temperatures $T_J$ and $T_B$. The black curve marks the previously determined g–DSM and the tuned parameter set g$^+$–DSM is shown in red. The inset displays $C/\chi$, and the solid line demonstrates that $T_{CW}(T) = 0$ at $T = T_J$

hence demonstrating that the temperature-dependent Curie–Weiss temperature $T_{CW}(T)$ equals zero at $T = T_J$. In the next section, we analyse the physical interpretation and consequences of these experimental results.

**Analogy to classical gases.** In this section, we explore the thermodynamic implications of a maximum in $\chi T/C$ and propose a strong analogy to the theory of classical gases. That there is a peak in $\chi T/C$ is not entirely surprising given that the sign of the effective nearest-neighbour interaction in Dy$_2$Ti$_2$O$_7$ is ferromagnetic, while the eventual expected ordering wave vector is most likely non-zero[15, 24]. What is more surprising is the 'peak temperature' where $\chi T/C$ reaches a maximum. It occurs at $T \approx 2$ K, or about 20 times the expected ordering temperature in Dy$_2$Ti$_2$O$_7$ (refs.[15, 24]), and a factor 2 or so above the well-studied peak temperature for the specific heat, which signals the rapid crossover from the paramagnetic regime to the spin ice state[7, 33–35]. The main aim of this study is to understand the temperature scale and physical origin of the peak in $\chi T/C$ shown in Fig. 1.

To start with, we consider the consequences of a peak in $\chi T/C$ by introducing the thermodynamic potential

$$F = S - U/T,$$ (2)

with a total differential

$$dF = -U d\left(\frac{1}{T}\right) - \left(\frac{1}{T}\right)\mu_0 V H_{int} dM.$$ (3)

Cross differentiating with respect to $M$ and $1/T$, we obtain the relation

$$\left(\frac{\partial U}{\partial M}\right)_T = \mu_0 V \left[H_{int} - T\left(\frac{\partial H_{int}}{\partial T}\right)_M\right],$$ (4)

which implies

$$\left(\frac{\partial U}{\partial M}\right)_T = 0 \rightarrow \chi + T\left(\frac{\partial \chi}{\partial T}\right)_M = 0 \rightarrow \frac{d(\chi T)}{dT} = 0.$$ (5)

We therefore find that an extremum in $\chi T/C$ implies $(\partial U/\partial M)_T = 0$. Similarly, it follows that the temperature-

dependent Curie–Weiss temperature, $T_{CW}(T)$ vanishes at the peak temperature, as shown in the inset of Fig. 1. That the internal energy, $U$, is independent of the magnetisation is a strong and intuitive definition of an effectively ideal non-interacting system. This is reminiscent of certain special conditions in gas thermodynamics, the best known defining the Boyle temperature, where the second virial coefficient vanishes and the ideal equation of state is obeyed. In fact, we see in Fig. 1 that there is a special temperature that corresponds to the Boyle temperature, namely the temperature at which $\chi T/C$ equals unity, at $T_B = 0.57$ K.

In the Methods section, we show in detail how $p/T$ for a gas or $H/T$ for a magnet may be expressed as the sum of the familiar ideal equation of state (ideal gas law or Curie law, respectively) plus a non-ideal term, that we label $q$. In both cases, the sign of the function $q$ reflects the sign of the net interaction in the system. Thus, for a gas, $q_{gas}$ is essentially the virial expansion: $q_{gas} = \sum_{i=2}^{\infty} B_i(T)(n/V)^{i-1}$, which for many purposes may be truncated at the second term ($i = 2$). In that case, $B_2$ is an integral over the pair potential $u_{ij}$, where the integrand depends on the Mayer function $(e^{-u_{ij}/kT} - 1)$. The sign of $q \propto B_2$ thus indicates the net interaction: positive for repulsive and negative for attractive. For the Van der Waals gas, $B_2 = b - a/RT$ and the net interaction switches sign precisely at the Boyle temperature $T_B = a/(bR)$, reflecting the crossover from net repulsion at high temperature to net attraction at low temperature. The critical and Joule–Thomson temperatures are determined by the same energy scale with numerical pre-factors 8/27 and 2, respectively. Similarly, for a magnet, $q_{mag} = \sum_{r\neq 0} \Gamma_r(T)$, where $\Gamma(\mathbf{r})$ is the pair correlation function. It is therefore positive for net ferromagnetic correlations (analogous to repulsive interactions in the gas as they tend to make make $M$ or $V$ larger) and negative for net antiferromagnetic ones (analogous to attractive interactions in the gas as they tend to make $M$ or $V$ smaller). Therefore, in both a gas and a magnet, the Boyle temperature, $T_B$, marks the temperature at which competing interactions cancel each other to give an apparently ideal equation of state.

We explore further thermodynamic analogies in the Methods section while here we simply summarise the main results in Table 1. In order to work out and understand the microscopic and phenomenological origin of these results, we begin by discussing the microscopic models used to describe spin ice in the next section.

**Spin ice models.** The interactions in spin ice materials stem from the ions with magnetic moments $\mathbf{\mu}_i$ which reside on the corners of the pyrochlore lattice of corner-sharing tetrahedra[36]. As a result of the nature of the crystal-field doublet ground state[37–40], the magnetic moments are Ising-like[40] and confined to point towards the centres of the adjacent tetrahedra. The

**Table 1 The special temperatures in magnets and gases**

| Temperature | Paramagnet | Gas |
|---|---|---|
| $T_B$ | $\chi T/C = 1$ | $pV/nRT = 1$ |
| $T_J$ | $\dfrac{d(\chi T/C)}{dT} = 0$ | $\left[\dfrac{\partial(pV/nRT)}{\partial T}\right]_{V/n} = 0$ |
| $T_{JT}$ | $\dfrac{d(\chi C/T)}{dT} = 0$ | $\left[\dfrac{\partial(pV/nRT)}{\partial T}\right]_p = 0$ |

Summary and comparison of the Boyle (B), Joule (J) and Joule–Thomson (JT) special temperatures for a magnet and a gas. The special temperatures listed are all indicators of quasi-ideality. For spin ice, $T_{JT}$ is infinite, while for the Van der Waals gas, $T_J$ is infinite

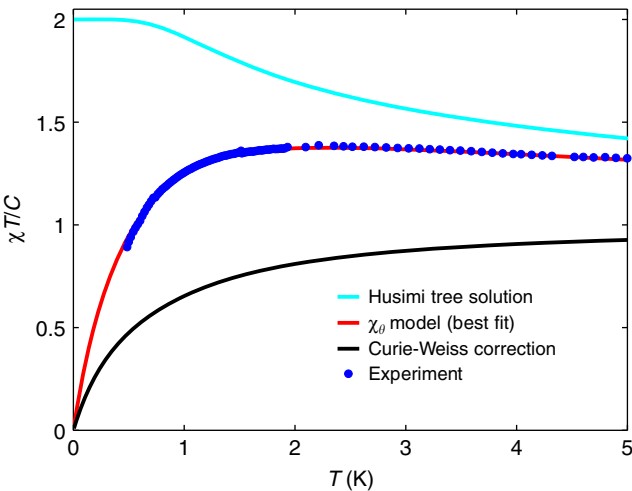

**Fig. 2** Phenomenological susceptibility model for the inverting magnet Dy$_2$Ti$_2$O$_7$. Husimi tree solution $\chi T/C$ for the NNSI model in cyan, the Curie–Weiss correction in black and their product, the best fit to the phenomenological $\chi_\theta$ model (Eq. (10)) in red. Experimental data for Dy$_2$Ti$_2$O$_7$ are shown as blue filled circles

primary magnetic interactions are the dipolar and short-range exchange interaction, and the materials are modelled by the dipolar spin ice model (DSM)

$$\mathcal{H} = J_1 \sum_{\langle i,j \rangle} \mathbf{S}_i \cdot \mathbf{S}_j + Da^3 \sum_{i>j} \frac{\mathbf{S}_i \cdot \mathbf{S}_j - 3\left(\hat{\mathbf{r}}_{ij} \cdot \mathbf{S}_i\right)\left(\hat{\mathbf{r}}_{ij} \cdot \mathbf{S}_j\right)}{r_{ij}^3}, \quad (6)$$

where $r_{ij}$ is the distance between spin $i$ and $j$, $D$ the dipolar interaction and $J_1$ the nearest-neighbour exchange interaction. With no dipolar interaction ($D = 0$) and only nearest-neighbour exchange, this model reduces to the nearest-neighbour spin-ice model (NNSI), which describes spin ice quantitatively well down to about 0.6 K[41]. The NNSI has a completely degenerate ground state and does not order. Together, dipolar and nearest-neighbour exchange interaction lead to the standard dipolar spin-ice model (s–DSM)[35]. The dipolar interaction weakly breaks the degeneracy of the NNSI and induces a transition to an ordered state at very low temperature. In addition to the nearest-neighbour exchange interactions $J_1$, the generalised spin ice model (g–DSM) contains second and third nearest-neighbour interactions $J_2$, $J_{3a}$ and $J_{3b}$. A set of parameter values were previously determined ($J_1 = 3.41$ K, $J_2 = -0.14$ K, $J_{3a} = J_{3b} = 0.025$ K), which models a number of experiments at a quantitative level[28].

Another model of high conceptual and physical importance that elegantly captures salient features of spin ice systems is the dumbbell model[26], obtained by replacing the point-like dipoles of the spin ice materials by dipoles of finite length. In this manner, the dipolar (◐) Hamiltonian can be written as the sum of the monopolar (◯) dumbbell model and quadrupolar (◇) corrections[26, 42],

$$\mathcal{H}^{◐} = \mathcal{H}^{◯} + \mathcal{H}^{◇}. \quad (7)$$

Having introduced the models commonly used to describe spin ice, we are now in a position to model the experimental susceptibility of Dy$_2$Ti$_2$O$_7$ reported in Fig. 1.

**Phenomenological susceptibility model.** In order to begin describing phenomenologically the experimental downturn in $\chi T/$

$C$, we use the Husimi tree solution,

$$\frac{\chi_0 T}{C} = \frac{2 + 2e^{2\beta J_{\text{eff}}}}{2 + e^{2\beta J_{\text{eff}}} + e^{-6\beta J_{\text{eff}}}}, \quad (8)$$

for the susceptibility of the NNSI[29] as our starting point. The nearest-neighbour interaction is here denoted by $J_{\text{eff}}$. We assume that there is an additive correction to the Helmholtz-free energy of the NNSI, $\mathcal{F}_0$, and that the correction is quadratic in the magnetisation $M$,

$$\mathcal{F} = \mathcal{F}_0 - \frac{\theta M^2}{2C}. \quad (9)$$

We take $\mathcal{F}_0$ to be the NNSI-free energy obtained on the pyrochlore cactus and $\theta$ is a coupling parameter. Differentiating twice with respect to $M$ yields the sought correction to $\chi_0$:

$$\frac{\chi_\theta T}{C} = \frac{\chi_0 T}{C} \cdot \frac{1}{1 - \theta \chi_0 / C}. \quad (10)$$

The resulting susceptibility, $\chi_\theta$, is thus a product of $\chi_0$ and a Curie–Weiss like susceptibility $(1 - \theta\chi_0/C)^{-1}$. This model contains two parameters ($\theta$ and $J_{\text{eff}}$). In Fig. 2, we show the best fit to experimental data ($\theta = -0.277$ K, $J_{\text{eff}} = 1.531$ K), along with the two separate factors of the product. It is clear that $\chi_\theta$ models the experimental data well. Furthermore, the peak in $\chi T/C$ arises from the product of the monotonically decreasing function $\chi_0$ and the monotonically increasing function, $(1 - \theta\chi_0/C)^{-1}$. Note that $\chi_0 T/C$ has a low-temperature plateau close to 2 extending out to a temperature of about 1 K. It follows that an infinitesimally small, but finite negative $\theta$ induces a peak in $\chi T/C$ at a temperature $O(1)$ K. This is the 'mechanism' behind the elevation of $T_J$ to a surprisingly high temperature by very weak interactions. This phenomenon is a main result of our study.

The physical origin of $\theta$ is perhaps most easily viewed as a mean-field-like correction arising from beyond nearest-neighbour interactions and the weak ordering tendencies of the dipolar interaction. It constitutes a mean-field correction to the (Husimi tree) mean-field construct, which apparently, works rather well. In the next section, we discuss the microscopic interpretation of the $\theta$-correction further.

Finally, we would like to point out that this framework bears a close resemblance to a demagnetising correction, where $\chi_0$ is the external and $\chi_\theta$ the internal susceptibility. By differentiating Eq. (9) only once, we obtain a relation for the magnetic fields in the two models,

$$\mathbf{H} = \mathbf{H}_0 - \frac{\theta}{C}\mathbf{M}, \quad (11)$$

which has exactly the same form as the definition of the demagnetising field with the demagnetising factor equal to $\theta/C$. The demagnetising transformation is a sensitive function of the demagnetising factor[22], and evidently a similar sensitivity arises in our phenomenological model.

**Microscopic susceptibility model.** In the present section, we wish to determine how well our measured susceptibility can be modelled at a microscopic level and to establish a connection between our phenomenological model of the previous section and the microscopic theory. We begin by modelling our experimental data using the g–DSM model[28].

As shown in Fig. 1, the g–DSM parameter set results in a susceptibility that overshoots our experimental result. We thus found it necessary to slightly adjust the third nearest-neighbour

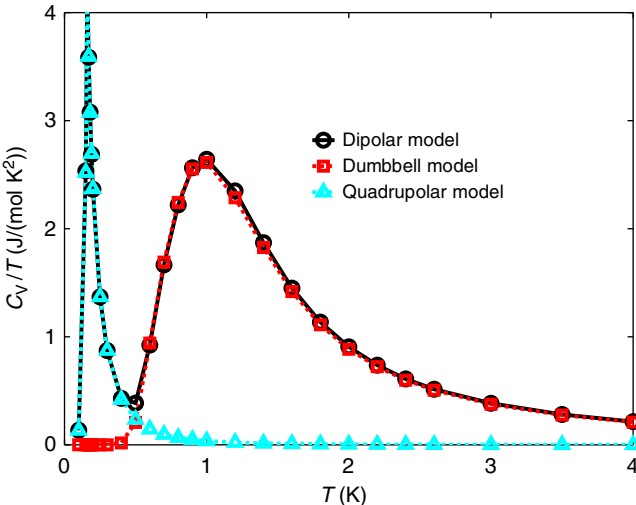

**Fig. 3** Dumbbell–quadrupolar decomposition of the spin ice specific heat. Specific heat for the g$^+$–DSM (black) along with the dumbbell (red) and quadrupolar (cyan) contributions

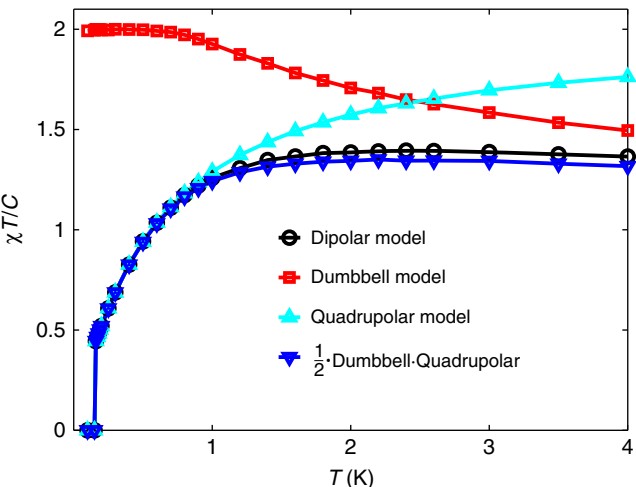

**Fig. 4** Dumbbell–quadrupolar decomposition of the spin ice susceptibility. Susceptibility $\chi T/C$ for the g$^+$–DSM (black) along with the dumbbell (red) and quadrupolar (cyan) contributions. The product of the dumbbell and quadrupolar susceptibility is shown in blue

parameter to $J_{3a} = 0.030$ K and $J_{3b} = 0.031$ K. As can be seen in Fig. 1, we obtain a close match to the experimental data with this new parameter set labelled g$^+$–DSM. We checked that such an adjustment of $J_{3a}$ and $J_{3b}$ leads to almost imperceptible changes in the neutron structure factor and the specific heat.

Having established a good microscopic model describing the experimental data, we would like to understand the connection between our phenomenological $\chi_\theta$ model and the microscopic g$^+$–DSM. In order to do this, we consider the monopolar–quadrupolar decomposition of the Hamiltonian, Eq. (7). The dumbbell model, $\mathcal{H}^{\bigcirc}$, like the NNSI, features perfectly degenerate spin ice states and a Curie crossover from $\chi T/C = 1$ at high temperature to $\chi T/C = 2$ at low temperature[29]. The quadrupolar model, $\mathcal{H}^{\diamondsuit}$, is short ranged, with interactions decaying as $1/r^5$ (ref.[26]). From Eq. (7), it follows that the energy of the ice states in the dipolar and quadrupolar model are equivalent, up to a constant shift and, therefore, we expect the short-range quadrupolar model to capture the low-temperature behaviour of spin ice. We have numerically decomposed the dipolar Hamiltonian and simulated these three models for finite systems. In Fig. 3, we show that the specific heat of the g$^+$–DSM model is indeed well described as the sum of a low-temperature part from the quadrupolar model restricted to the ice states and a high-temperature part calculated from the dumbbell model.

The connection to our phenomenological model $\chi_\theta$ follows from the following observations: Since $\lim_{T\to 0} \chi^{\bigcirc} = \lim_{T\to\infty} \chi^{\diamondsuit} = 2$, and the low (high) temperature behaviour of $\chi^{\lozenge}$ is well described by $\chi^{\diamondsuit}$ ($\chi^{\bigcirc}$), it follows that, to a good approximation,

$$\chi^{\lozenge} \approx \frac{1}{2}\chi^{\bigcirc} \cdot \chi^{\diamondsuit}. \tag{12}$$

We therefore note that the mean-field-like correction $(1 - \theta\chi_0/C)^{-1}$ to the Husimi susceptibility is closely related to the susceptibility of the quadrupolar corrections to the dumbbell model, $\chi^{\bigcirc}/2$, restricted to the ice states. We show in Fig. 4 that this is indeed a good approximation.

Finally, we note that since the dumbbell model does not order at any finite temperature, it should correspond to our phenomenological model with $\theta = 0$. However, a second

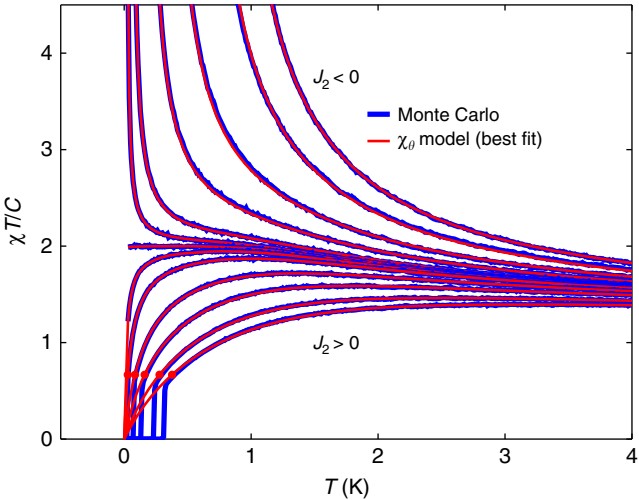

**Fig. 5** Modelling the spin ice susceptibility through the point of complete frustration. Tuning the dumbbell model from a ferromagnetic ($J_2 < 0$) to an antiferromagnetic ($J_2 > 0$) ground state. The $J_2$ range extends from $-0.2$ K for the uppermost curve to $+0.2$ K for the lowest curve. Monte Carlo data (blue) lie below the corresponding optimal two-parameter fit (red) to the phenomenological $\chi_\theta$ model (Eq. (10)). On the antiferromagnetic side, the solid red circles indicate $\chi_\theta T/C|_{T=-\theta} \approx 2/3$, which lies very close to the critical temperature in Monte Carlo, visible as vertical blue lines

nearest-neighbour exchange interaction induces a finite-temperature transition, which is ferromagnetic for $J_2 < 0$ and antiferromagnetic for $J_2 > 0$. Numerical access to the dumbbell model therefore allows us to check how well our phenomenological model captures the transition from an antiferromagnetic to a ferromagnetic ground state as we tune an additional second nearest-neighbour $J_2$. As can be seen in Fig. 5, the phenomenological model follows the tuned dumbbell model very closely right through the point of complete frustration ($J_2 = \theta = 0$). On the ferromagnetic side, the phenomenological parameter $\theta$ equals the critical temperature, $T_c = \theta$. On the antiferromagnetic side, one finds that $\lim_{T\to 0}\chi_\theta T/C|_{T=-\theta} = 2/3$, and in Fig. 5, we see that the ordering transition in the Monte Carlo simulations

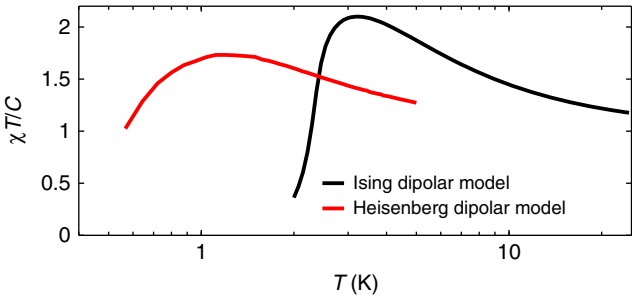

**Fig. 6** Susceptibility for the dipolar Ising and Heisenberg models on the cubic lattice. Monte Carlo calculation of $\chi T/C$ for the dipolar Ising (1000 spins) and Heisenberg (216 spins) models described by Eq. (6) with $J_1 = 0$, $Da^3 = 1$ and $\mathbf{S}_i = \pm\hat{\mathbf{z}}$ (Ising)

occurs when $\chi_\theta T/C \approx 2/3$. This relation can therefore be a useful experimental criterion as to when to expect an ordering transition. It also provides further evidence that the phenomenological model $\chi_\theta$ captures relevant physical aspects of frustrated ferromagnets.

To summarise, in this section we have thus shown that the experimental susceptibility is well matched by the g–DSM model with slightly adjusted third nearest-neighbour parameters (g$^+$–DSM). This shows that $\chi T/C$ provides access to interaction parameters that are otherwise hard to access. In addition, we have demonstrated that our phenomenological model is a good description of the microscopic dipolar model for a wide range of parameters, and that the phenomenological correction term describing the phase transition arises from the quadrupolar corrections to the dumbbell model.

## Discussion

If spin ice were the only inverting ferromagnet, the notion of special temperatures could be just a curiosity of limited interest. However, we have identified a number of compounds which feature a peak in $\chi T/C$. Kapellasite, a proposed quantum spin-liquid[12], is formed of kagome planes and features competing FM and AF interactions. There is a clear peak in $\chi T/C$ which, as in the case of Dy$_2$Ti$_2$O$_7$, hints at an eventual AF ordering. $\chi T/C$ for the quantum pyrochlore material Nd$_2$Zr$_2$O$_7$ increases as $T$ is lowered, but the peak is apparently pre-empted by a phase transition to an ordered all-in-all-out state[43–45]. Through Monte Carlo simulations, we have also verified that the well-studied Ising and Heisenberg models of classical spins coupled through dipolar interactions on the cubic lattice features a peak in $\chi T/C$, as shown in Fig. 6. The spinel GeCo$_2$O$_4$ also belongs to this class of magnets[46]. Finally, $\chi T/C$ peaks for a number of spin-glass materials such as the organic $\kappa$-(BEDT-TTF)$_2$Hg(SCN)$_2$Br compound[16] and Eu$_x$Sr$_{1-x}$S$_y$Se$_{1-y}$ (ref.[47]).

We have therefore demonstrated that there exists a class of inverting frustrated ferromagnets, which feature special temperatures at which the intrinsic competing FM and AF interactions balance and the magnets are quasi-ideal. At $T_B$, the magnetic Boyle temperature, the ideal equation of state is obeyed, and at $T_J$, the magnetic Joule temperature, the internal energy is independent of the magnetisation. Below $T_J$, the AF interactions start to dominate, and the corresponding peak in $\chi T/C$ is an indication of eventual AF order, barring further disruptive low-temperature terms in the Hamiltonian. Since the peak can occur at a high temperature relative to the eventual ordering temperature, it is a useful diagnostic feature in the quest for quantum and classical spin liquids. In a true spin-liquid, the competing FM and AF interactions should be delicately balanced so that there is no

finite Joule temperature, see Fig. 5. In our case study of Dy$_2$Ti$_2$O$_7$, the peak in $\chi T/C$ is caused by weak (quadrupolar) perturbations to the primary (monopolar) Hamiltonian, and provides a way to experimentally probe these corrections.

In this context, we also note that a common way to characterise the level of frustration in magnetic systems with strongly competing interactions is the frustration index $f \equiv T_{CW}/T_c$, see ref.[48], where $T_{CW}$ is the Curie–Weiss temperature and $T_c$ the critical temperature. However, there are many systems for which this measure is not suitable, such as low-dimensional systems for which $T_c = 0$, or systems with either strongly anisotropic components of the g tensor or highly anisotropic exchange[49]. The field of highly frustrated magnetism would thus benefit from other indicators of operating high frustration, which relies on ratios of temperature scales that are readily experimentally available, such as the Joule temperature for inverting magnets.

By introducing the concept of special temperatures in frustrated ferromagnets, we have filled a notable gap in the well-established thermodynamic analogy between magnets and classical gases. While our present investigation has focused on systems featuring a maximum in $\chi T/C$, we note that the converse phenomenon occurs in, for example, ferrimagnetic spin chains[50]. In these systems, antiferromagnetic correlations at high temperature cross over to an eventual low-temperature ferromagnet. The prevalence of such behaviour is a question we leave for future investigations.

## Methods

**Susceptibility measurement**. The magnetic susceptibility was measured using a Quantum Design SQUID magnetometer and the crystals were positioned in a cylindrical plastic tube to ensure a uniform magnetic environment. Measurements were performed in the reciprocating sample option operating mode to achieve better sensitivity by eliminating low-frequency noise. The position of the sample was carefully optimised to minimise misalignment with respect to the applied magnetic field. In particular, the sphere was measured at different positions and orientations in order to confirm the isotropic response and to fully reproduce the results of ref.[30].

Low-temperature magnetic susceptibility was measured using a Quantum Design MPMS SQUID magnetometer equipped with an $i$Quantum $^3$He insert[51]. In analogy with ref.[30], different measurements were made: low-field susceptibility (at $\mu_0 H_0 = 0.005$, 0.01 and 0.02 T) and field-cooled (FC) versus zero-field-cooled (ZFC) susceptibility. Also, magnetic field sweeps at fixed temperature were performed in order to evaluate the susceptibility accurately and confirm the linear approximation. The FC versus ZFC susceptibility measurements involved cooling the sample to base temperature 0.5 K in zero field, applying the weak magnetic field, measuring the susceptibility while warming up to 2 K, cooling to base temperature again and finally re-measuring the susceptibility while warming. Before switching the magnetic field off, field scans with small steps were performed in order to estimate the absolute susceptibilities.

To increase the statistics and control for dynamical effects, three measurements were taken at each temperature before warming to the next step point. Furthermore, to test the accuracy of the measurement, some data were acquired with an increased number of raw data points—typically 64 points rather than the usual 24. In fact, at low temperature, the magnetic moment of the sample is close to the saturation value of the instrument, especially at $\mu_0 H_0 = 0.02$ T. When measuring under such conditions, it is necessary to increase the number of raw data points to 64 to maintain consistency between measurements.

Data have been compared with the high-temperature measurements described in ref.[30], in particular in the overlapping region $1.8 \leq T \leq 2$ K. Without further manipulation, the two sets of data are in very good agreement with variations of the order of <0.5%. This can be attributed to the uncertainty in the actual field value in each of the two instruments, mainly due to the presence of small frozen fields in the superconducting coils. In Fig. 1, the two sets of measurements were accurately superimposed, by compensating (<0.5%) the actual applied field value of the low-temperature measurement.

**Magnetic thermodynamics**. To make an analogy between magnetic and fluid thermodynamics, we define $X$ and $x$ as the extensive and intensive mechanical variable, respectively. For a fluid, $X = V$ (volume) and $x = p$ (pressure) and we assume the mole number $n$ is fixed. For a magnet, we assume an ellipsoidal sample and deal with intrinsic properties (post-demagnetising correction). We have $x = H$, the internal $H$-field and $X = -\mu_0 VM$, where $M$ is the magnetisation. Here, the minus sign is included to complete the analogy, but it makes no difference to the following results.

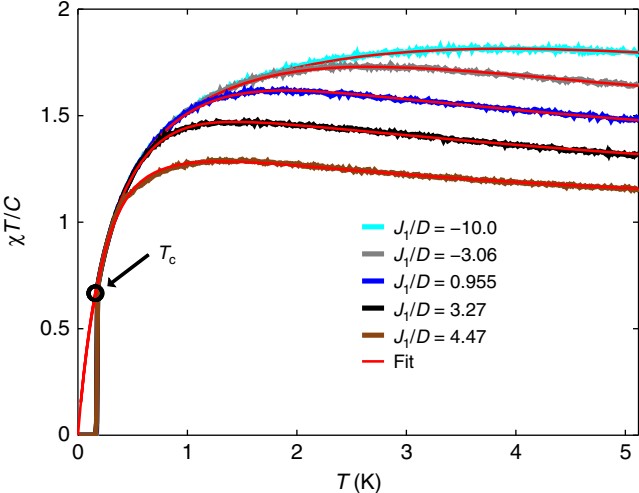

**Fig. 7** Phenomenological susceptibility for the standard dipolar spin ice model. Susceptibility $\chi T/C$ of s–DSM as a function of temperature $T$ and $J_1/D$ ratio. Monte Carlo data lies below the corresponding optimal two-parameter fit (red) to the phenomenological $\chi_\theta$ model (Eq. (10)). The arrow indicates $T_c$ in the Monte Carlo simulation, which corresponds well to $\chi_\theta T/C|_{T=-\theta} \approx 2/3$, marked by the black circle

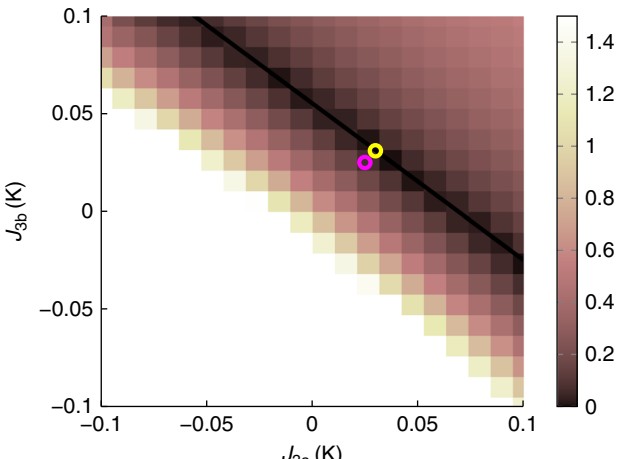

**Fig. 8** Fine-tuning the third nearest-neighbour exchange interactions. $\chi T/C$ RMS-deviation of the DSM with $J_1 = 3.41$ K and $J_2 = -0.14$ K compared to experiments. The purple ring is the g–DSM[28] and the yellow ring the g$^+$–DSM (this work). The black line is a guide for the eye of the minimum RMS

A strong and intuitive definition of an ideal non-interacting system is that the internal energy depends only on temperature: $(\partial U/\partial X)_T = 0$. This implies $x - T(\partial x/\partial T)_X = 0$ as an equivalent definition of ideality. Integration of the latter then shows that the non-interacting equation of state is of the form:

$$\frac{x}{T} = \phi(X). \quad [\text{non} - \text{interacting}] \tag{13}$$

where $\phi$ is some function. Indeed, this is true for both the ideal gas and the ideal paramagnet, where the functions in question are $\phi_{mag} = M/C$, where $C$ is the Curie constant and $\phi_{gas} = nR/V$, where $R$ is the gas constant. For a real gas or paramagnet, we write the equation of state as:

$$\frac{x}{T} = \phi(X) + q(X, T), \quad [\text{real}] \tag{14}$$

where the function $q$ is the non-ideal correction. We now define the following special temperatures: these may not be unique, but we will refer to them in the singular for clarity.

The Boyle temperature, $T_B$, is defined as the temperature where $q = 0$, so the ideal equation of state happens to be obeyed.

The Joule temperature, $T_J$, is the temperature where $(\partial U/\partial X)_T = 0 \Rightarrow x - T(\partial x/\partial T)_X = 0$, which indicates that the intensive variable $p$ or $H$ is tangentially proportional to absolute temperature $T$. This temperature is infinite for a Van der Waals gas, but may be finite for some real gases (e.g., helium) and some magnets.

The Joule–Thomson temperature, $T_{JT}$, is defined as the temperature where: $(\partial x/\partial T)_E = 0 \Rightarrow X - T(\partial X/\partial T)_x = 0$. where $E = U + xX$ is the enthalpy. For a typical gas, $T_{JT}$ is finite at any density, and in this sense, a real gas never reaches the ideal gas limit. For the magnetic models considered here, $T_{JT}$ is infinite.

We can see that $T_B$ and $T_J$ indicate quasi-ideality, where some criteria of ideality are satisfied. The third special temperature, $T_{JT}$, indicates that $X \propto T$ tangentially. This corresponds to quasi-ideality only in the particular case of a gas ($V \propto T$) and not in the case of a magnet ($M \propto T$). Nevertheless, there is a symmetry between $T_J$ and $T_{JT}$: both are defined by setting to zero a Legendre transform $\mathcal{L}[z] = z - T(\partial z/\partial T)$ of the intensive variable $z \to x$ and the extensive variable $z \to X$, respectively. Hence, both imply tangential linearity of the corresponding variable with absolute temperature.

Starting with these Legendre transforms, we can translate the three special temperatures into conditions on the function $T\phi(X)/x$. We are interested in the magnet in the linear regime at low field and magnetisation, where we can define the susceptibility $\chi = M/H$, which is a function of $T$ only: $\chi = \chi(T)$. Hence, for a magnet, we obtain conditions on $\chi T/C$ which, in this context, is analogous to $pV/nRT$. The Boyle temperatures, $T_B$, are located by $\chi T/C = 1$ and $pV/nRT = 1$. The Joule temperature, $T_J$, for a magnet corresponds to an extremum in $\chi T/C$ as a function of temperature. The Joule–Thomson temperature, $T_{JT}$, for a gas at fixed pressure corresponds to an extremum in $pV/nRT$. These relations are summarised in Table 1.

**Applicability of the phenomenological model.** In order to establish the applicability of the phenomenological $\chi_\theta$ model, Eq. (10), we compare it here to the standard dipolar spin ice model (s–DSM), Eq. (6), which includes the dipolar interaction $D$ in addition to a nearest-neighbour exchange interaction, $J_1$. In this model, spin ice behaviour persists up to $J_1/D < 6.01$, and the dipolar interaction induces a low-temperature phase transition to a 'single-chain' state[15, 24]. The model features a corresponding Joule temperature, and as can be seen in Fig. 7, our phenomenological model describes the susceptibility of the s–DSM remarkably well down to, and including, the critical temperature $T_c$, at which $\chi_\theta T/C|_{T=-\theta} \approx 2/3$.

**Determination of model parameters.** The parameters for the g$^+$–DSM were chosen in the following manner: $J_1 = 3.41$ K and $J_2 = -0.14$ K were set to the previously determined values of the g–DSM[28]. Then, a $\chi T/C$ RMS chart was calculated for the deviation between our experimental data and Monte Carlo calculations as a function of $J_{3a}$ and $J_{3b}$ (see Fig. 8). From the chart, we determined the point closest to the g–DSM values ($J_{3a} = J_{3b} = 0.025$ K) located in the minimum RMS valley. This point is indicated by a yellow ring in Fig. 8, corresponding to the g$^+$–DSM values ($J_{3a} = 0.030$ K, $J_{3b} = 0.031$ K).

**Code availability.** The custom computer codes used in this study are available from the corresponding author on reasonable request.

**Data availability.** The data sets generated and analysed in this study are available from the corresponding author on reasonable request.

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

## Acknowledgements

We thank D. Prabhakaran for providing crystals from which the samples were cut. The idea to numerically determine the quadrupole correction to the dumbbell model is due to S. Powell[42]. We are grateful to Addison Richards for providing Monte Carlo simulation data for the dipolar Heisenberg model on a cubic lattice. S.T.B. thanks J. Xu and B. Lake for a useful correspondence concerning $Nd_2Zr_2O_7$. We thank Wen Jin for pointing out ref.[50] to us. The simulations were performed on resources provided by the Swedish National Infrastructure for Computing (SNIC) at the Centre for High Performance Computing (PDC) at the Royal Institute of Technology (KTH). M.T. and P.H. are supported by the Swedish Research Council (2013-03968), M.T. is grateful for funding from Stiftelsen Olle Engkvist Byggmästare (2014/807), and L.B. is supported by The Leverhulme Trust through the Early Career Fellowship programme (ECF2014-284). The work at the University of Waterloo was supported by the Canada Research Chair programme (M.J.P.G., Tier 1). This research was supported in part by the Perimeter Institute for Theoretical Physics. Research at the Perimeter Institute is supported by the Government of Canada through Innovation, Science, and Economic Development Canada and by the Province of Ontario through the Ministry of Research, Innovation, and Science.

## Author contributions

L.B. performed all high-temperature measurements and data collection. L.B. and O.A.P. performed the low-temperature SQUID measurements. M.T. and P.H. performed all simulations and the dumbbell–quadrupole decomposition. S.T.B. conceived the phenomenological model and the concept of special temperatures. M.J.P.G. realised the broad applicability of the concept of inverting magnets. T.F. contributed to data analysis. L.B., M.T., M.J.P.G., S.T.B. and P.H. wrote the manuscript with input from all authors.

## Additional information

**Competing interests:** The authors declare no competing interests.

