## [Peer Review File · Nature Communications]

REVIEWERS' COMMENTS:

Reviewer #1 (Remarks to the Author):

The manuscript under review makes quite an interesting connection between thermodynamics of the non-ideal gas and that of frustrated ferromagnet. The connection, the essence of which is summarized in eq.5 and Table 1, leads to magnetic analogues of the Boyle, Joule and Joule-Thomson special temperatures. It also suggests a new useful parameter, $\chi T/C$, maximum of which (as a function of temperature T) represents the "Joule" temperature at which the internal energy U does not depend on magnetization M .

I find this consideration to be new and relevant, and am in favor of publication of this work in Nature Communications.

I do have one simple suggestion: it does appear to me that making yet stronger connection with the second virial coefficient B of the real gas would actually be quite useful. As is well known from the classical thermodynamics, that coefficient is determined by the integral of the Mayer function. This fact is responsible for $B(T)$ changing sign upon cooling from high to low temperature T . In the magnet, that coefficient is determined by the integral of spin correlation function -- and therefore too goes to zero at some special temperature at which contributions from ferromagnetic and antiferromagnetic interactions exactly compensate each other. The authors do make this connection in the Methods section (Joule-Thomson temperature). In my opinion, this nice physical argument should be expanded upon and moved into the main part of the manuscript.

Reviewer #2 (Remarks to the Author):

In this Manuscript the authors establish a theoretical analogies between frustrated, correlated spin systems of mixed ferro and antiferro-magnetic interaction and the theory of gases where special temperatures signal ideal states. The authors explore first the case of rare earth pyrochlore where the competition between ferro and antiferromagnetic couplings comes from an admixture of dipolar and exchange interactions as well as from the anisotropic nature of the dipolar interaction at further neighbors. There they perform lower than before measures of magnetic susceptibility while using state of the art crystals. These measurements alone would be of broad relevance.

Indeed there they find a peak in $\chi T/C$ which implies stationarity of the internal energy with respect to magnetization: that is suggestive of an ideal, non interacting state at the corresponding temperature.

Further, they investigate the origin of magnetic special temperatures via a simplified model to enucleate the relevant mechanisms. They note that the nearest neighbor approximation for spin ice fails to reproduce the experimental data, but can be corrected by the addition of a terms depending on further nearest neighbors, where the dipolar interaction being anisotropic changes sign. To this phenomenological model they seek validation via different alteration of microscopic models which include both dipolar integration and further neighbors exchange couplings. While their tinkering with models might appear to be slightly ad hoc, the relevant point here is not to find a perfect model for the phenomenon. Rather it is to notice that only models with sufficient fidelity in dealing with further neighbors interactions can reproduce special temperature features, strongly suggesting that the shape of the $\chi T/C$ lends itself as a probe of delicate, competing couplings, often neglected in analysis.

It seems to be that this signature is more general than spin ice systems and could be a feature of other systems of coupling with competing signs.

The ms is clearly and well written.

Because of its potential broad impact, of the novelty and relevance of the new measurements and of the intelligent analysis I do recommend publication as it is.

Reviewer #3 (Remarks to the Author):

I recommend the paper by Bovo et al. entitled "Special temperatures in frustrated ferromagnets" for publication in Nature Communications.

The paper shows high precision susceptibility measurements on a spherical sample of Dy₂Ti₂O₇ down to 0.5 K and compares them to values for dipolar spin model using Monte Carlo simulations. The authors then compare the characteristic temperatures of this spin system with the characteristic temperatures in gases. Such a comparison between magnetic systems and gases is to my knowledge is unprecedented and of general interest, as it manages to map the complicated interactions in frustrated magnets to equivalent parameters in a gas. Because Dy₂Ti₂O₇ has such a well characterized Hamiltonian, the authors are able to formulate a phenomenological model for the magnetic susceptibility. This model exposes the mechanism which induces the special temperatures and explained how minute frustrated exchange interactions lead to effects at surprisingly high temperatures. The paper is clearly written and easily accessible and is not only limited to specialists in the field. Given the general interest in frustrated magnets, it will be well received.

RE: NCOMMS-18-02995A

Special temperatures in frustrated ferromagnets

by L. Bovo, M. Twengström, O.A. Petrenko, *et al.*

REVIEWERS' COMMENTS:

Reviewer #1 (Remarks to the Author):

The manuscript under review makes quite an interesting connection between thermodynamics of the non-ideal gas and that of frustrated ferromagnet. The connection, the essence of which is summarized in eq.5 and Table 1, leads to magnetic analogues of the Boyle, Joule and Joule-Thomson special temperatures. It also suggests a new useful parameter, $\chi T/C$, maximum of which (as a function of temperature T) represents the "Joule" temperature at which the internal energy U does not depend on magnetization M .

I find this consideration to be new and relevant, and am in favor of publication of this work in Nature Communications.

I do have one simple suggestion: it does appear to me that making yet stronger connection with the second virial coefficient B of the real gas would actually be quite useful. As is well known from the classical thermodynamics, that coefficient is determined by the integral of the Mayer function. This fact is responsible for $B(T)$ changing sign upon cooling from high to low temperature T . In the magnet, that coefficient is determined by the integral of spin correlation function -- and therefore too goes to zero at some special temperature at which contributions from ferromagnetic and antiferromagnetic interactions exactly compensate each other. The authors do make this connection in the Methods section (Joule-Thomson temperature). In my opinion, this nice physical argument should be expanded upon and moved into the main part of the manuscript.

We thank the Referee for reading our manuscript and for his/her support for publication. We have followed the referee's suggestion by moving and adjusting the discussion on the virial coefficient to the subsection "Special temperatures - analogy to classical gases." from the methods section.

Reviewer #2 (Remarks to the Author):

In this Manuscript the authors establish a theoretical analogies between frustrated, correlated spin systems of mixed ferro and antiferromagnetic interaction and the theory of gases where special temperatures signal ideal states. The authors explore first the case of rare earth pyrochlore where the competition between ferro and antiferromagnetic couplings comes from an admixture of dipolar and exchange interactions as well as from the anisotropic nature of the dipolar interaction at further neighbors. There they perform lower than before measurements of magnetic susceptibility while using state of the art crystals. These measurements alone would be of broad relevance.

Indeed there they find a peak in $\chi T/C$ which implies stationarity of the internal energy with respect to magnetization: that is suggestive of an ideal, non interacting state at the corresponding temperature.

Further, they investigate the origin of magnetic special temperatures via a simplified model to enucleate the relevant mechanisms. They note that the nearest neighbor approximation for spin ice fails to reproduce the experimental data, but can be corrected by the addition of a terms depending on further nearest neighbors, where the dipolar interaction being anisotropic changes sign. To this phenomenological model they seek validation via different alteration of microscopic models which include both dipolar interaction and further neighbors exchange couplings. While their tinkering with models might appear to be slightly ad hoc, the relevant point here is not to find a perfect model for the phenomenon. Rather it is to notice that only models with sufficient fidelity in dealing with further neighbors interactions can reproduce special temperature features, strongly suggesting that the shape of the $\chi T/C$ lends itself as a probe of delicate, competing couplings, often neglected in

analysis.

It seems to be that this signature is more general than spin ice systems and could be a feature of other systems of coupling with competing signs.

The ms is clearly and well written.

Because of its potential broad impact, of the novelty and relevance of the new measurements and of the intelligent analysis I do recommend publication as it is.

We thank the Referee for reading our manuscript and for his/her support for publication. We have taken no action in regard to the report from Reviewer #2.

Reviewer #3 (Remarks to the Author):

I recommend the paper by Bovo et al. entitled "Special temperatures in frustrated ferromagnets" for publication in Nature Communications.

The paper shows high precision susceptibility measurements on a spherical sample of Dy₂Ti₂O₇ down to 0.5 K and compares them to values for dipolar spin model using Monte Carlo simulations. The authors then compare the characteristic temperatures of this spin system with the characteristic temperatures in gases. Such a comparison between magnetic systems and gases is to my knowledge is unprecedented and of general interest, as it manages to map the complicated interactions in frustrated magnets to equivalent parameters in a gas. Because Dy₂Ti₂O₇ has such a well characterized Hamiltonian, the authors are able to formulate a phenomenological model for the magnetic susceptibility. This model exposes the mechanism which induces the special temperatures and explained how minute frustrated exchange interactions lead to effects at surprisingly high temperatures. The paper is clearly written and easily accessible and is not only limited to specialists in the field. Given the general interest in frustrated magnets, it will be well received.

We thank the Referee for reading our manuscript and for his/her support for publication. We have taken no action in regard to the report from Reviewer #3.